# "My home is (now) at peace": Evaluating the relevance, acceptability and potential scalability of a guided self-help intervention for male refugees in Uganda

**Jacqueline N. Ndlovu**[1]*, **Lena S. Andersen**[1], **Marx R. Leku**[2], **Nawaraj Upadhaya**[1,2], **Morten Skovdal**[3], **Jura L. Augustinavicius**[4‡], **Wietse A. Tol**[1,2,5,6‡]

**1** Global Health Section, University of Copenhagen, Copenhagen, Denmark, **2** HealthRight International, New York, United States of America, **3** Section of Health Services Research, University of Copenhagen, Copenhagen, Denmark, **4** Department of Equity, Ethics, and Policy, School of Population and Global Health, Faculty of Medicine and Health Sciences, McGill University, Montreal, Canada, **5** Athena Research Institute, Vrije Universiteit Amsterdam, Amsterdam, Netherlands, **6** Arq International, Diemen, Netherlands

☉ These authors contributed equally to this work.
‡ JLA and WAT are joint senior authors on this work.
* Jacqueline.ndlovu@sund.ku.dk

## Abstract

In humanitarian crises, male refugees face significant mental health challenges, including high rates of depression and alcohol misuse. However, access to adequate mental health services is limited. In this study, we evaluate a combined intervention that addresses both mental health and alcohol misuse among male refugees in Uganda, focusing on its relevance, acceptability, and potential for scalability. We conducted a qualitative study, using process evaluation data, to evaluate a combined guided self-help intervention for mental health and alcohol use among male refugees in Uganda's Rhino and Imvepi camp refugee settlements. We used thematic network analysis to identify themes related to relevance, acceptability, and potential scalability. 28 in-depth process evaluation interviews were conducted in total. Participants included male South Sudanese refugees who had received enhance usual care, or SH+ only or SH+ and ASSIST-BI combined. Family members of the participants and intervention facilitators were also interviewed. Results highlighted the intervention's relevance, emphasising the need for participants to be providers and productive citizens. Acceptability was underscored by a sense of community and social acceptance, particularly evident in SH+ group sessions. Facilitators noted that the intervention format was key to scalability, despite barriers such as competing priorities and resource limitations. These results highlight the importance of addressing mental health and alcohol misuse simultaneously, demonstrating the combined intervention's relevance, acceptability and scalability. We emphasise the need for a comprehensive approach that integrates additional support mechanisms, such as livelihoods, to enhance overall impact while preserving the interventions core components. This broader understanding is important for developing effective and sustainable solutions in similar humanitarian contexts.

**Data availability statement:** All data generated or analysed during this study are included in this published manuscript (except for interview transcripts due to confidentiality).

**Funding:** This project is funded by Research for Health in Humanitarian Crises (R2HC), a division of Elhra, grant number #32396. The funders had no role in study design, data collection and analysis, decision to publish, or preparation of the manuscript. NU, JA and WT were part of the project funded by R2HC.

**Competing interests:** The authors have declared that no competing interests exist.

## Background

There is a high burden of mental health challenges among populations affected by armed conflicts, with one in five people experiencing mental health problems [1]. Mental health challenges negatively affect quality of life, physical health, and the ability to function [2]. In humanitarian settings, male refugees often face a disproportionately high burden of psychological distress, coupled with substance misuse [3]. The experience of forced displacement, exposure to violence, loss of social support networks, and general uncertainty exacerbates these challenges [4]. In many cases, psychological distress—such as anxiety, depression, and trauma—leads to alcohol misuse as a coping mechanism, providing temporary relief. However, this misuse worsens mental health issues, creating a vicious cycle of emotional distress and impaired functioning [5]. This bidirectional relationship is particularly evident in refugee populations, where displacement and the loss of social support increase vulnerability to both conditions[6]. Alcohol misuse, while offering short-term escape, worsens these mental health issues and reinforces harmful behaviours, making it a maladaptive strategy with long-term detrimental effects[7].

Alcohol is often used as a coping mechanism to relieve stress from adapting to new environments and to manage mental health conditions such as traumatic stress, anxiety, and depression [8]. Among displaced populations, alcohol misuse is particularly common, offering temporary relief from these psychological challenges [9]. Displacement can leave men feeling helpless and socially isolated, which often drives them to alcohol as an escape [10]. However, while it may offer short-term relief, alcohol misuse worsens mental health issues and reinforces harmful behaviours, making it a maladaptive strategy with long-term detrimental effects.

Given the harmful cycle between psychological distress and alcohol misuse, addressing both issues together is crucial for developing effective interventions. Therefore, there is an urgent need for targeted interventions that tackle both the mental health and substance use challenges faced by male refugees in humanitarian settings [11]. Despite the clear need for interventions, access to mental health services remains limited in humanitarian settings. Most settings lack adequate mental health facilities, trained professionals, and specialized services to meet the complex needs of male refugees [12]. Scalable interventions are essential in these resource-constrained contexts. Task shifting, where non-specialists deliver mental health care, offers a promising solution [13]. By enabling briefly trained individuals to provide mental health support, task shifting bridges the gap in services, making care more accessible and sustainable in under-resourced areas [14].

Although approximately 75% of refugees are hosted in low- and middle-income countries, most studies on alcohol misuse and mental health have been conducted in high-income settings [15]. These studies also tend to examine alcohol misuse and mental health challenges separately, despite the well-established interrelationship between the two [16–18]. This gap in the evidence highlights the need for research focused on refugee populations in low- and middle-income countries, such as Uganda, where this study was conducted.

This study focuses on male South Sudanese refugees living in northern Uganda, where displacement has created significant challenges. Traditionally, South Sudanese men are viewed as family providers and protectors, but unemployment and reliance on humanitarian aid have disrupted these roles, leading to a loss of identity, heightened psychological distress, and increased substance misuse[19]. The cultural expectation of masculinity intensifies these struggles, as many men face the burden of being unable to fulfill their traditional roles [19]. Additionally, cultural differences between refugees and the Ugandan host community, along with stigma surrounding mental health, exacerbate feelings of isolation and contribute to worsening outcomes [19].

In this study, we examine a combined intervention that addresses both mental health challenges and alcohol misuse, using process evaluation data. We conducted a process evaluation

as part of a mixed methods, feasibility cluster randomised controlled trial [fCRT] of an intervention that combined components to address psychological distress [i.e., Self Help Plus, or SH+] and alcohol misuse [i.e., the Alcohol, Smoking and Substance Involvement Screening Test and Brief Intervention, or ASSIST BI] with male refugees in Uganda. The original fCRT study aimed to assess the feasibility and acceptability of both the intervention and research protocols. The purpose of this study is to:

(i)  Evaluate the *relevance* and *acceptability* of the combined intervention [addressing psychological distress and alcohol misuse] and study procedures

(ii) Understand perceptions of factors that may either support or inhibit the *scalability* of the combined intervention

In this study, we refer to *relevance* as the appropriateness, practicability and perceived usefulness of the intervention [20]. *Acceptability* refers to satisfaction with intervention components, content and delivery [20], and *potential for scale* up refers to the potential to provide the intervention to larger groups of people [after initial piloting and effectiveness testing in rigorous evaluation studies]. By combining interventions that address both mental health and substance misuse, and conducting a process evaluation to assess implementation, this study aims to contribute valuable insights into scalable mental health care for refugee populations in low-resource settings.

## Methods

We used a qualitative study design to evaluate the relevance, acceptability and potential scalability of a combined version of a guided self-help intervention addressing mental health and alcohol use in male adult refugees in Rhino Camp and Imvepi refugee settlements, Uganda. Data used in this study was collected as part of process evaluation activities conducted. Ethical clearance was obtained from the Mildmay Uganda Research Ethics Committee [MUREC REC REF 0406-2020] and all participants provided written informed consent for participation, administered and witnessed by trained research assistants.

### Setting

This study was conducted in two refugee settlements in northern Uganda: Rhino Camp and Imvepi refugee settlement. Rhino Camp refugee settlement currently hosts about 140 000 refugees and continues to welcome new arrivals [21]. In comparison, Imvepi refugee settlement hosts about 65 000 refugees [22]. Male refugees in Rhino Camp refugee settlement tend to be highly mobile, as they move between their host country and their countries of origin depending on fluctuations of unrest and livelihoods opportunities. On the other hand, male refugees in Imvepi refugee settlement are not as mobile, as the settlement is semi-permanent. Over 90% of refugees living in both settlements are from South Sudan. Around 60% of the total population living in the settlements are refugees, with 40% consisting of people from host communities. The main source of livelihood among the refugees is subsistence farming and about 20% of refugees reported having an occupation in 2022 such as casual labour [23]. Some of the challenges faced by the refugees in this region include poverty, conflicts attributed to gender-based violence for example, and economic vulnerability [24].

### Participants

This study included participants who were part of the fCRT. The fCRT consisted of three study arms, namely; [i] enhanced usual care where participants were provided with an information session from a community psychosocial assistant about emotional problems

and alcohol/substance use, and how to seek help for these challenges, [ii] participants in an SH+ only arm and [iii] participants who were in the SH+/ASSIST-BI arm, i.e., the combined intervention.

## Inclusivity in global research

Additional information regarding the ethical, cultural, and scientific considerations specific to inclusivity in global research is included in the Supporting Information [S1 Checklist].

## Recruitment and sampling

In the overall project, a total of 214 participants were screened, with 181 participants meeting the inclusion criteria [South Sudanese male, speaking Juba Arabic, 18 years and older, with at least moderate psychological distress or moderate alcohol and substance usage]. Men with severe alcohol use and/or low psychological distress, with psychosis and at imminent risk of suicide were excluded and referred to services that could assist them better. To assess risky alcohol and other drug use, an 8-item screening tool called the ASSIST was used. To assess psychological distress, the Kessler 6 [K6] was used. Moderate psychological distress was classified as greater than or equal to a K6 total score of 5 [≥5] as in previous studies with South Sudanese refugees [25,26] and others [27,28]. Recruitment began on the 1st of May, 2022 and was completed on the 30th of June, 2022.

For this study, participants who were enrolled in the fCRT were approached for participation using a purposive sampling approach. We used purposive sampling to ensure the inclusion of participants with specific characteristics relevant to our study's objectives, thereby enhancing the depth and relevance of the findings [29]. We selected participants who met the following criteria for process evaluation interviews:

(i)     Participants with moderate risk of substance use who completed ASSIST-BI and SH+

(ii)    Participants with moderate risk of substance use who completed ASSIST-BI but did not complete SH+

(iii)   Participants with moderate risk of substance use who did not complete ASSIST-BI or SH+

(iv)    Participants with low risk of substance use who completed SH+

(v)     Participants with low risk of substance use who did not complete SH+

(vi)    Family members of the participants

(vii)   ASSIST-BI and SH+ facilitators

In this study, completion was defined as attending all ASSIST-BI sessions and at least four of the five SH+ sessions.

## Interventions

**Self help plus [SH+].**  This study focuses on a guided self-help intervention called SH+. SH+ was developed by the World Health Organization [WHO] as a potentially scalable strategy for reducing psychological distress and managing adversity more broadly [30]. Based on acceptance and commitment therapy [ACT], SH+ consists of an illustrated self-help book and an audio recording. SH+ is provided in workshops [20-30 participants] over a period of five weeks, with one session lasting up to 90 minutes per week. SH+ can be delivered by briefly trained non-specialists through task shifting approaches. In this study,

task shifting involved training non-specialists to deliver SH+ and ASSIST-BI to participants. This approach increases access to mental health services in resource-limited settings. The impact on participants' experiences can vary; some may feel more comfortable with familiar, community-based facilitators, while others may perceive non-specialist care as less formal or authoritative. SH+ has been previously translated, adapted and implemented in Uganda [including in Juba-Arabic for refugee women], Turkey and five western European countries [Italy, Germany, Austria, United Kingdom and Finland] [31–34]. These studies demonstrated clinically meaningful improvements in mental health outcomes, and SH+ was found to be safe, effective, acceptable, feasible and relevant across diverse socio-cultural settings [32,33]. Among female refugees in Uganda, SH+ effectively reduced psychological distress [25]; helped prevent mental health concerns [26], and supported psychological flexibility, the hypothesized working ingredient of the intervention [35]. Initial piloting of the guided self-help intervention with male refugees in 2015 showed that separate adaptations may be necessary, particularly with regard to additional alcohol use concerns [31]. Due to these different needs, this study focuses on the link between mental health and alcohol misuse among male refugees, which led to the development of a combined intervention that addresses both challenges.

**Alcohol, smoking and substance involvement screening test [ASSIST] and brief intervention [BI] [ASSIST-BI].** To address alcohol misuse, we used WHO's ASSIST-BI [36]. The package focuses on how best to manage the challenges of substance use in clinical and non-specialist health care settings, such as community settings. It consists of two components: the screening component [ASSIST] and the intervention component [ASSIST BI]. Screening allows the scoring of patterns of problems related to substance misuse through a questionnaire and has cut off points to categorise harmful use and dependence. This is followed up by a brief intervention component to practically help people facing substance misuse challenges. The intervention is a brief, easy to administer procedure based on motivational interviewing principles [37]. In low and middle income countries [LMIC]s, few studies have evaluated the efficacy of MHPSS interventions in reducing harmful alcohol misuse [38]. A key limitation in existing studies is that outcome measures of the interventions have not been validated among the target impact populations, and/or interventions are too brief to support efficacy [38].

**Combined SH+ ASSIST-BI.** To combine SH+ to include ASSIST-BI elements, the research team first developed a flipbook and a handout for the participants. These materials were developed to supplement the current participant handbook, which requires a high level of literacy that the target impact group did not necessarily possess. The ASSIST-BI flipbook and handout were then rigorously translated from English to Juba Arabic. A local illustrator helped in creating images for the flipbook and the handout, to ensure that participants [including those who were illiterate] could understand the materials well. The relevance, comprehensibility, and acceptability of the content was further explored through interviews with healthcare workers and lay persons. In the process of training intervention facilitators, feedback was collected on the materials and adaptations were made based on the suggestions made, such as including context specific exercises and examples.

## Procedures

At the end of the intervention period, a series of in-depth process evaluation interviews were conducted with participants, their family members and intervention facilitators. These were conducted to better understand their experiences with the combined SH+ and ASSIST-BI intervention. The topic guide was based on the UK Medical Research Council [MRC] framework for process evaluations of complex interventions [39]. Topic guides covered themes such as general demographic factors, contextual factors affecting implementation, implementation processes and mechanisms of impact. Process evaluation interviews were conducted by

trained research assistants from June to July 2022. A total of 28 interviews were conducted by trained research assistants with men who had been part of the feasibility trial, men's family members, and intervention facilitators. Participants were recruited based on their availability and willingness to participate in the process evaluation interviews. Interviews with men who had been part of the feasibility trial and their family members were conducted in Juba Arabic and then translated and transcribed in English by the research assistants. Interviews with intervention facilitators were conducted in English. All interviewees consented to being audio recorded and audio recordings were transcribed verbatim.

## Data analysis

In this study, we explored three areas: relevance, acceptability and potential scalability. We used thematic network analysis to analyse data by identifying patterns and categorizing these patterns into themes, following the steps outlined by Attride-Sterling [40]. Thematic network analysis is a valuable qualitative research approach for handling complex data by identifying overarching themes and their relationships [40]. We used this approach as it offers a structured way for exploring patterns and it enables transparent communication of findings. Initial codes were developed deductively, taking inspiration from topic guide themes. Topic guide themes were broadly focusing on overall impressions, specific experiences on the different skills and materials, intervention attendance, involvement of family and friends and helpfulness of intervention skills and materials. One of the authors [JNN] developed a coding framework using the dataset. This coding framework was then applied to the rest of the dataset and used to code the data in NVivo 14 by JNN [41]. Once all interview transcripts were coded, hierarchies of the codes were created based on the code count. Codes were then clustered into broader categories or basic themes based on similarities and relationships. These basic themes were then evaluated on the basis of the aim of the study and research objectives. We identified 26 themes that directly contributed to our overall analysis. We then explored relationships between these themes and clustered them together into six organising themes. The organising themes spoke directly to the global themes; relevance, acceptability and potential for scale up, which our research objectives focus on. We did not apply any theory at this stage. Participants were assigned pseudonyms to further protect their identity.

## Results

Twenty-eight process evaluation interviews were conducted as part of the study [Table 1]. The participants of this study are categorised into three; [i] intervention participants, [ii] family members of intervention participants, and [iii] intervention facilitators. Most intervention participants from the combined intervention [who completed both SH+ and ASSIST-BI sessions] participated in the process evaluation. Completion of SH+ sessions refer to having attended four sessions or all five sessions. Only one intervention participant included in the study had completed enhanced usual care [EUC]. Most of the family members of the intervention participants included were spouses, and the rest were either a parent of the intervention participant or their child. Participants in the intervention were all male South Sudanese refugees, with the youngest being 23 years of age and the oldest being 64 years old. The family members of the participants were also from South Sudan. Intervention facilitators where mostly from Uganda and had been selected as facilitators based on proficiency in Juba Arabic. Information on overall participant characteristics and demographics is displayed in Table 1.

This results section comprises of eight main topics: [i] need to be a provider, [ii] strategies used to cope with adversity, [iii] thinking too much, [iv] self-reported change in behaviour

**Table 1. Sociodemographic information of participants.**

| Characteristic | Male [N/%] | Female [N/%] | Total [N/%] |
|---|---|---|---|
| **Intervention participants** | | | |
| Completed both SH+ and ASSIST-BI sessions | 8 [29%] | 0 [0%] | 18 [65%] |
| Completed ASSIST-BI sessions but not SH+ | 2 [7%] | 0 [0%] | |
| Completed EUC | 1 [4%] | 0 [0%] | |
| Completed SH+ sessions | 4 [14%] | 0 [0%] | |
| Completed 3 or 4 SH+ sessions | 3 [11%] | 0 [0%] | |
| **Family members of intervention participants** | | | |
| Parent [mother/father] | 1 [4%] | 1 [4%] | 6 [21%] |
| Spouse [wife] | 0 [0%] | 3 [11%] | |
| Children | 1 [4%] | 0 [0%] | |
| **Intervention facilitators** | | | |
| Facilitators | 0 [0%] | 4 [14%] | 4 [14%] |

post intervention, [v] sense of community building social acceptance, [vi] facilitators during implementation, [vii] barriers during implementation and [viii] recommendations to improve scalability. Parts [i] to [iv] are related to *relevance of the intervention*, part [v] is related to *acceptability of the intervention* and parts [vi] to [viii] are related to *potential for scalability of the intervention*. More detailed information on the themes is illustrated in Table 2. While some of the themes in Table 2, such as community building, relationships, and social acceptance, could be interpreted as aspects of relevance, these elements were integral to participants' satisfaction with the social aspects of the intervention. Therefore, they were categorized under acceptability as they significantly influenced participants' overall experience and satisfaction with both the content and delivery of the intervention. Similarities and differences between participants from the SH+ only arm and the SH+/ASSIST-BI arm are described, as we were comparing across these categories.

## Relevance of the intervention

**Need to be a household provider and a productive citizen.** The need to be a provider for the family and to be a community member who contributes positively to society were consistently mentioned as a key motivator for either reducing drinking or stopping drinking completely, across participants in the SH+ only arm and SH+/ASSIST arm:

> "I feel responsible, I was able to care more about my family and manage them, also be helpful to my community and those around me." Alfred, aged 49, completed 3 or 4 SH+ sessions

This was also echoed among family members, who witnessed how the intervention had brought about a change in terms of instilling a sense of responsibility and caretaker attitude to their loved ones:

> "I am positive about the program because it made him a brand-new man; he started helping the family with the basic needs like food, paying [the] children's school [fees] and catering for our medical bills." Emma, aged 39, family member [spouse]

Some of the men who were part of the intervention also mentioned how their drinking activities made them feel undervalued and despised by their families and communities. In

**Table 2. Thematic network of relevance, acceptability and potential for scalability of the intervention.**

| Relevant research objective | Organising themes | Emerging themes |
|---|---|---|
| **Relevance of the intervention** | Need to be a household provider and a productive citizen | Money challenges<br>Poverty<br>Being helpful<br>Being valued |
| | Strategies used to cope with adversity | Coping strategies<br>Forget problems<br>Substance use<br>Reasons for participation<br>Alcohol use |
| | Thinking too much | Influence from friends<br>Conflicts<br>Past trauma<br>Conflicts |
| | Self-reported changes in behaviour post intervention | Reduced intake<br>Quit alcohol<br>Problem management<br>Stress management<br>Keeping busy |
| **Acceptability of the intervention** | Sense of community building and social acceptance | Relationships<br>Sharing problems<br>Socializing with people<br>Togetherness<br>Knowledge sharing |
| **Potential for scalability of the intervention** | Facilitators [what worked well during implementation] | Stakeholder engagement<br>Intervention format |
| | Barriers [what could have been improved during implementation] | Competing priorities<br>Inadequate resources and time |
| | Recommendations [what could be done to improve scalability] | Stakeholder engagement<br>Training and supervision<br>Funding and resources<br>Motivation |

those cases, the men recognized that when they drank too much, they disrupted other people's lives and, in some cases, became violent, causing negative reactions from relatives and community members. However, having an intervention that targeted them specifically as men was seen to bring back their value, and the men began the process of perceiving themselves as worthy again:

> "Some of us were traumatized and others were waiting to die but since death is from God, with the intervention of HealthRight, we felt we still have value." Ben, aged 64, completed all SH+ sessions

For the men who were part of the control arm and only received the enhanced usual care [EUC], they did not find the intervention as helpful. One of the men in particular shared that his friends who he used to drink with had changed positively after the intervention and were more responsible. However, although he had been advised on coping strategies and ways to reduce harmful alcohol consumption, he still struggled with drinking too much:

> "I learnt the few tips on how to reduce or quit drugs which I feel is not helpful because I could have learnt a lot if I was part of the program but unfortunately I was left out of it." Chris, aged 42, completed EUC

Participants in the SH+ only arm also reported a strong need to provide for their families. However, the need to be a contributor to society was reported to a lesser extent. The participants in this arm focused more on using the skills from the SH+ sessions to manage their emotions, to focus on the present and to ensure that they were contributed more at a family level:

"It is through these trainings that my home is at peace and that I am an important person to myself and the people around me." Daniel, aged 32, completed all SH+ sessions

**Strategies used to cope with adversity.** Participants from both the SH+ only arm and the SH+/ASSIST-BI arm reported various ways in which they had previously tried to cope with the difficulties they face. They mentioned drinking too much or using drugs and other substances which gave them temporary relief from 'thinking too much' and helped them to go to sleep. Additionally, participants also noted that being part of either the SH+ or SH+/ASSIST-BI intervention had made them realise that their previous coping strategies were not only harmful to themselves but were also harmful to their families as well. SH+/ASSIST-BI participants talked about how they used to use the little money that they got to fund their drinking habits, instead of taking care of their families. Also, they mentioned realising that the time spent in the trading centres with friends drinking could have been time spent working in the fields and providing food and an income for their families after being part of the SH+/ASSIST intervention. Furthermore, SH+/ASSIST-BI participants mentioned how the skills and strategies that they had learned from the intervention had helped them to find better ways of coping and avoiding risky situations:

"I stay with friends who don't drink and it teaches me good behaviours since they can stay peacefully [with others] without looking for problems." Ethan, aged 27, completed SH+ and ASSIST-BI sessions

SH+/ASSIST-BI participants relied more on weighing negative and positive aspects of drinking too much to help them decide on whether to quick alcohol or to reduce harmful alcohol consumption. They also drew from SH+ sessions for strategies on managing emotions and coping with adversity, which were some of the reasons that contributed to harmful alcohol consumption. On the other hand, SH+ participants reported different ways in which they now coped with adversity. They primarily used strategies from SH+ like grounding exercises to let go of the past and focus more on the present, sharing problems and acting on their values. They also used the SH+ book at home to remind them of the teachings they had received during SH+ sessions when they were going through a particularly challenging time:

"I use it [the SH+ book] especially in situations of boredom and idleness. It helps you get back to the present situation, it helps you cope up with urges and changes in life which makes you to adapt and accept your present state of mind." Alfred, aged 49, completed 3 or 4 SH+ sessions

**Thinking too much.** Some of the challenges that the participants shared that led them to thinking too much involved the stark differences between their lives when they were in South Sudan and their current lives in Uganda. This was a shared sentiment across both SH+ only and SH+/ASSIST-BI participants. The participants shared that they had previously been capable of taking care of their families and providing for their children. The participants also

shared that in South Sudan, they had been gainfully employed and had other leisure activities that kept them busy and well entertained:

"I had my business back home in South Sudan and as result of the war, many of my properties were lost." George, aged 32, completed all SH+ sessions

In contrast, the men found life in Uganda very different. One of the major challenges that the participants shared was how they were now idle and did not have the energy to take up their previous hobbies. They shared that the more time they had on their hands, the more intrusive thoughts such as feeling like a failure for not being able to provide as a man should, would come. The participants also talked about losing family and friends in the conflict in South Sudan and about feelings of guilt over their own activities during the conflict which contributed to thinking too much.

"Yes, I have learnt a lot from the intervention because I was a victim of suicide because of over thinking which resulted into stress but through the skills I learnt from the teaching I can handle my own problems, help my neighbours and friends when they are overwhelmed with thoughts." Henry, aged 44, completed all SH+ sessions

The participants mentioned that they would choose to drink as drinking gave them relief when they were being overwhelmed by thoughts. However, drinking too much would also lead to bad decisions that had even worse repercussions.

**Self-reported changes in behaviour post intervention.** Both SH+ and SH+/ASSIST-BI participants reported that they had either reduced their alcohol consumption or completely quit drinking alcohol after being part of either SH+ only or SH+/ASSIST-BI intervention. About half of the participants in both study arms reported having either quit alcohol or reduced on their alcohol consumption. The way to reduce alcohol consumption varied across the participants, with some saying that they no longer drank daily but chose to only drink over the weekends, or when there were big celebrations in the community. Others reported drinking smaller quantities per day, as compared to what they had previously been drinking before the intervention:

"I want to tell you that a lot has changed because I used to take alcohol extremely throughout the day but trust me that has completely changed. I now take a very small quantity of alcohol and not on daily basis." James, aged 26, completed SH+ and ASSIST-BI sessions

Some participants with multiple substance use challenges reported that being part of SH+/ ASSIST-BI not only helped them to manage their alcohol consumption, but also helped them to manage intake of other substances as well. SH+ participants on the other hand seemed to consume alcohol primarily and there were fewer reports of other substance misuse in this group. Family members also confirmed the self-reported behaviour changes that participants shared, and added their own personal experiences as people who had intimate knowledge and proximity on a daily basis with participants:

"He used to use a lot of drugs like marungi and cigarettes, sometimes he would even drink but when he started attending those teachings, he has completely changed, he doesn't use any drugs or alcohol." Sarah, aged 20, family member [spouse]

In addition to reducing alcohol consumption, half of the participants from both the SH+ intervention and the ASSIST-BI intervention shared that they had opted to stop drinking

alcohol completely. Among the reasons cited, knowledge about the negative effects that alcohol has on the health of a person, and how drinking excessively affects relationships were mentioned the most. Participants across both interventions also mentioned that the intervention sessions helped them to realize that some of their problems, which they thought were impossible to solve, were exacerbated by excessive drinking of alcohol. For example, most participants talked about how they always quarrelled with their partners, causing them to avoid being at home and opting to spend most of their time drinking with friends. They only realized the vicious cycle they were in during the sessions, and some of the participants made a decision then to quit drinking completely:

"I have decided not to drink alcohol because it helps me not to have quarrel and fight with my wife." Jack, aged 55, completed SH+ and ASSIST-BI sessions

For other participants, they initially committed to reducing their alcohol consumption as they felt that stopping drinking completely was too much of a drastic change. However, as the sessions progressed and they learnt more about the effects of drinking alcohol and other ways of managing distress and problems, they then decided to stop drinking alcohol altogether:

"After I got exposed to the health risks, I decided to quit drinking and using drugs because I realised the negative implications it had in my life. I first reduced my alcohol use before I eventually quit it." Sam, aged 63, completed SH+ and ASSIST-BI sessions

Family members reported on some of their own observed changes in behaviour. For example, one family member talked about how his father was now able to support their family after stopping to drink and using his time to gain skills that could provide an income:

"Yes, he has left drinking alcohol completely and he has joined a vocational training where he has learnt carpentry and building skills from which he is supporting the family now." Nathan, aged 22, family member [son]

From the participants and family members perspectives, both SH+ and SH+/ASSIST-BI interventions helped men who had been excessively drinking reduce their alcohol consumption or stop drinking completely. As most of the reasons for drinking alcohol excessively were linked to distress, ways of coping with problems, and filling in time as the men were less busy, the intervention sessions were found to be helpful in addressing these challenges:

"Through this program of SH+, my father was able to leave drinking alcohol and he was able to develop positive thoughts which made him to learn the skills of carpentry and building by joining a vocational training just because of the courage he got from the facilitator." Nathan, aged 22, family member [son]

With all the positive changes in behaviour that the participants and their family members experienced, the perceived need for the intervention to benefit those that had not had the chance to be involved in the intervention, was highlighted by most participants. Some of the participants mentioned that they struggled with maintaining former friendships with the people that they used to drink with. The participants also wanted these friends to have access to the SH+ and SH+/ASSIST-BI interventions and the same help that they had received. Moreover, being idle had previously been identified as another contributor to excessive drinking of alcohol. Participants who were in the SH+ intervention felt that although the intervention had helped them in different ways, there was still a concern related to idleness which participants

felt needed to be addressed more in the SH+ sessions. Reducing alcohol consumption or stopping completely meant that the participants now had more time on their hands. More time meant more thinking too much for the participants, which they felt could contribute to relapsing. Thus, they expressed the need for the intervention to also include other activities that could keep them busy. This was also echoed by SH+/ASSIST-BI participants and family members as well:

> "The organisation should involve them [participants] in income generating activities so that they are not idle but busy to avoid stress." Rachel, aged 46, family member [mother]

### Acceptability of the intervention

**Sense of community building social acceptance.** Excessive drinking of alcohol came with a varied number of challenges among participants. Although not highlighted by all participants, most participants across both the SH+ intervention and SH+/ASSIST-BI intervention mentioned feeling isolated from the rest of their families and communities who do not drink alcohol. Their social circle consisted only of the friends that they drank alcohol with, which made them spend more and more time together. Other participants chose to isolate themselves as drinking alcohol was not part of their culture and was perceived in a negative manner by society. In those cases, the participants chose to drink in isolation and hide from the rest of the community. One of the participants who had previously isolated himself reflected on this:

> "[The intervention] relieves me from alcohol thoughts and gives me freedom to socialize with people because alcohol is taken in hidden places away from people." Ethan, aged 27, completed SH+ and ASSIST-BI sessions

Family members also experienced having someone in their family who drank alcohol isolating themselves, and being overall unhappy when around family. In addition to this, the family members also talked about how they chose to isolate a member of the family who drank a lot, particularly when that individual became aggressive and abusive when drunk. However, after the intervention, participants felt that they had become less disruptive and problematic in their families and communities. For some, the behaviour change was quite evident, and they were recognised and rewarded by others for their efforts in changing their behaviour for the better:

> "His relationship with the neighbours and the community has improved and he has been chosen as the community leader in this village." Anna, aged 25, family member [spouse]

The sense of improved community relationships that accompanied either reducing or quitting alcohol was echoed across participants and family members alike. Family members felt that their family units were being re-built and likened it to when they were living peacefully in South Sudan. Other participants simply appreciated being valued and liked in the community:

> "The community now likes me because I do not cause problems like when I used to drink alcohol." Henry, aged 44, completed all SH+ sessions

Going beyond the family, participants who had been part of the SH+ intervention, either in the SH+ only group or in the SH+/ASSIST-BI group, appreciated meeting with other men in

the community who were having similar challenges during the SH+ group sessions. This gave the participants a sense of community and a safe space where they could share their problems and hear from other men in similar situations. With a deeper understanding and having had the opportunity to develop feelings of empathy, participants noted an improved relationship with other people in the community:

> "What made me more interested in the group session was that, other members shared their experiences, they taught us to love one another, and help each other in times of need. Additionally, this improved relationships among the people in the same community." Oliver, aged 44, completed SH+ and ASSIST-BI sessions

### Potential for scalability of the intervention

**Facilitators [what worked well during implementation].**  Intervention facilitators reported on several factors that they perceived worked well during implementation, and that could be continuously replicated in other settings as well as during scaling up. These included the group format for SH+ sessions, which was well liked by the participants and enabled an increase in the reach of SH+ in a short period of time. The language used during the SH+ and SH+/ASSIST-BI interventions [Juba Arabic] was also considered a facilitator for implementation:

> "The language used during the sessions favoured everyone since it is a common language spoken by all South Sudanese." Sophia, aged 31, intervention facilitator

Another important facilitator of implementation that the intervention facilitators high-lighted was the investment in building and maintaining a strong relationship with other stake-holders in the settlement, such as community leaders and church leaders. This helped in the easier mobilisation of participants and also built trust among stakeholders as their community leaders communicated about the interventions prior to implementation.

**Barriers [what could have been improved during implementation].**  Competing priorities in the settlement among participants were perceived to be a barrier to implementation and subsequent scale-up, as reported by intervention facilitators. The intervention facilitators reported that in some cases, they had to reschedule sessions or sessions that were not well attended because participants were otherwise engaged in other activities:

> "Another challenge that made it hard for the participants was that, most of them were engaged in activities with other organizations. The same participants would be taken up somewhere and this made it hard for them to come [for SH+ and SH+/ASSIST-BI sessions]." Olivia, aged 29, intervention facilitator

These were often activities with other partners in the settlement or livelihood activities such as farming. For the intervention facilitators themselves, they reported that the biggest barrier to implementation and potential scale-up was limited resources to motivate them or to support the addition and training of other intervention facilitators to reduce their workload.

**Recommendations [what could be done to improve scalability].**  Intervention facilitators offered several recommendations for future scale-up activities based on their experience of being part of the intervention. As one of the biggest barriers experienced during implementation involved participants having competing priorities, intervention facilitators

recommended closely liaising with other settlement stakeholders to coordinate activities and stay up-to-date with other settlement activities, such as food distribution days. They also recommended that intervention facilitators should have refresher trainings during the implementation process to upskill them and ensure that quality is maintained throughout implementation. Intervention facilitators also recommended that supervision was more effective when done face-to-face, as opposed to through online supervision. Overall, the intervention facilitators recommended increasing funds for project activities that could support staff motivation and aid in staff retention, while also increasing organisational capacity to support scale up activities.

> "More funds and [intervention] facilitators to be allocated since the current [intervention] facilitators are few and the areas covered is small." Olivia, aged 29, intervention facilitator

## Discussion

Our study evaluated whether a guided self-help intervention alone, or a guided self-help intervention combined with an intervention to address alcohol misuse, were perceived to be relevant, acceptable, and potentially scalable among male South Sudanese refugees in Uganda. Based on the perspectives of participants, their family members, and intervention facilitators, the intervention was found to be highly relevant and acceptable in this setting for men who drank alcohol excessively and experienced psychological distress.

The combined intervention, SH+/ASSIST-BI, was perceived to be more relevant than SH+ only, as it directly addressed harmful alcohol and substance misuse. Participants in the combined intervention, SH+/ASSIST-BI, used strategies from both interventions to reduce and/or quit alcohol use. Based on the findings from this study, the combined intervention should have had a larger impact as compared to SH+ alone. However, perceived change in behaviour related to either quitting alcohol or reducing alcohol intake was interestingly similar across both interventions in this study. Half of the participants in each intervention self-reported quitting alcohol, while the other half self-reported reducing their alcohol intake. In another study, combination social protection [where both social protection interventions such as cash transfers are provided together with psychosocial care] showed strong HIV prevention effects among adolescents in South Africa [42]. This indicates that combination social protection may be potentially more effective than single interventions [43]. We also specifically aimed to understand the different perspectives on scalability potential of the combined intervention. The intervention was found to be relevant and acceptable and participants identified a great need to also reach more people in similar circumstances. However, this study identified no obvious reported differences in potential for scalability between SH+ and the combined intervention, SH+/ASSIST-BI.

While discussing alcohol misuse and psychological distress, participants also highlighted the stigma and discrimination they faced within their communities and from their families. For many, excessive drinking was associated with social isolation, either self-imposed or as a result of being isolated by family and community members. Participants shared that drinking alcohol was perceived negatively in society, leading them to hide their behaviour or drink in isolation. This stigma compounded their mental health challenges and created additional barriers to seeking support. The combined intervention played a key role among participants to overcome this social stigma, rebuild their relationships, and reintegrate into their communities.

There have been growing calls in the MHPSS field to scale up interventions through multisectoral integration, where mental health and psychosocial services are integrated with

services from other sectors. In the recent research priorities for MHPSS in humanitarian settings, integration of mental health into other sectors is ranked quite high [top five of the 20 prioritised research agendas] [44]. In our study, participants highlighted a gap that was not addressed by either the SH+ intervention or the SH+/ASSIST-BI intervention. The interventions examined in this study only addresses mental health or mental health and alcohol use issues. The relevance of the interventions could have potentially been improved by integrating SH+ and SH+/ASSIST-BI with livelihood activities to more holistically address the day-to-day challenges that the participants face that lead them to drink alcohol and to reduce chances of relapse. Including other elements such as livelihood activities could potentially improve psychological distress and foster economic wellbeing – two integral aspects linked to both alcohol and other substance misuse.

There have been investigations demonstrating that stand-alone livelihood interventions can promote good mental health and wellbeing [45]. The working hypotheses in studies such as these are that poverty and unemployment are two of the biggest stressors that affect the mental health and wellbeing of refugees. As much as this holds true, refugees also face other stressors that affect their mental health and wellbeing, and that cannot be addressed by economic empowerment alone. One particular study found that the impact of an integrated MHPSS and livelihoods intervention was much larger as compared to a standalone livelihoods intervention [46]. There are also other studies that have successfully integrated MHPSS interventions with interventions from other sectors [47–49]. For example, one study found that livelihood interventions may improve mental health and help people living with HIV better integrate with their communities [50]. Thus, integrating an MHPSS intervention such as SH+ with a livelihoods component and a component to address alcohol misuse is more likely to address mental health and substance misuse outcomes and their social determinants in a holistic manner.

Another important finding from our study points towards masculinity and what it means to the men to be a provider. From our findings, the men's role in the family was primarily to provide for one's family. When the men were unable to fulfil that role, they felt like they had failed and perceived themselves as "less-than". From a cultural perspective, the majority of South Sudanese communities adhere to patriarchal values [51]. Men dominate social, cultural and political decision making, which is consistent with norms in other neighbouring countries in the region [52]. However, it is also important to note that not all South Sudanese communities or families are inclined towards these social norms. For the communities that subscribe to these gender roles, men experience a culture shock when they are not able to fulfil their roles, or when their roles are replaced and women become providers instead. Growing flexibility in gender roles and norms is not unique to the South Sudanese refugees in Uganda [53]. Over the last decades, there has been more transition in these social norms as a consequence of globalisation for example, but in some communities such as the ones involved in this study, cultural norms remain strong. The results of this study illustrate how mental health challenges associated with moving from South Sudan to Uganda, and the losses incurred during this traumatic process led to the men being unable to fulfil their roles as per their norm. With the intervention, the men who changed their behaviour for the better by reducing and stopping their consumption of alcohol felt that their value had been instilled back in them. The men were able to use their time to seek employment opportunities and were better able to provide for their families. This brought back balance in family dynamics, and the safe and familiar cultural context of gender norms.

While our findings point towards the importance of being connected to the family and the community during recovery, there are tensions in understanding the concept of community for men who engage with friends when drinking. In some cases, drinking alcohol was an

activity conducted alone, linked to shame, loneliness and isolation. But, in other cases, the men created social circles where they spent most of their time with friends drinking alcohol. In those cases, when the men decided to quit or to reduce drinking, their friends were the greatest influencers for relapse. Although the men could practice refusal skills, they also experienced a sense of loss of a community that they had been a part of for a long time. Thus, it would be beneficial and more sustainable in terms of long-term impact to recruit groups of male friends together and to foster a shared goal that strengthens their social circles instead of causing isolation.

This study not only uncovers the factors contributing to the success of integrated psychological interventions but also sheds light on the complex life circumstances and predicaments faced by refugee men. Through an in-depth exploration of their experiences, the study provides valuable insights into the social, economic, and mental challenges that worsen their susceptibilities. By understanding these underlying issues, we can better appreciate the multifaceted nature of their challenges and the critical need for holistic, culturally sensitive and combined intervention strategies that address both their mental health and socio-economic needs.

A strength of our study is that it was able to highlight participants' and other stakeholders' own perspectives. We find it important to discuss limitations as well. Firstly, more participants in the SH+/ASSIST intervention arm were included in the process evaluation. This was primarily due to convenience sampling and the participants in that study arm were more willing to be recruited for process evaluation interviews. Thus, our results are not necessarily a reflection of all participants, as we could have included more participants from the control arm who received EUC and participants who dropped out as well. Secondly, only one person involved in implementation of the intervention was also involved in the study team [team responsible for the research] analysing the results. We might have had a better understanding of the results if more people in the study team were also involved in the implementation activities. However, it could also be said that because of the separation between the implementation and the study, we were more objective in reflecting on the findings as we did not have prior interaction with project activities. Thirdly, including a quantitative perspective could have triangulated with the qualitative self-reported behaviour change on alcohol consumption and mental health states after the SH+ and SH+/ASSIST-BI interventions.

## Conclusion

In summary, our study sheds light on the benefits of addressing mental health concurrently with addressing alcohol misuse among South Sudanese refugees in Uganda in a manner that is relevant, acceptable and has potential for scaling up. A broader understanding of how to incorporate other elements that can provide an income, such as livelihoods, while maintaining the integrity of the intervention is needed.

## Supporting information

**S1 Checklist. Inclusivity in global research.**
(DOCX)

## Acknowledgments

The authors extend their sincere gratitude to the South Sudanese men and their families for their invaluable participation. Special thanks are also due to the intervention facilitators, research assistants, and the dedicated staff at HealthRight for their significant contributions to this research.

## Author contributions

**Conceptualization:** Jacqueline Ntombizodwa Ndlovu, Lena S. Andersen, Morten Skovdal, Jura L. Augustinavicius, Wietse A. Tol.

**Data curation:** Jacqueline Ntombizodwa Ndlovu, Marx R. Leku.

**Formal analysis:** Jacqueline Ntombizodwa Ndlovu.

**Funding acquisition:** Nawaraj Upadhaya, Jura L. Augustinavicius, Wietse A. Tol.

**Investigation:** Lena S. Andersen, Marx R. Leku, Nawaraj Upadhaya, Jura L. Augustinavicius, Wietse A. Tol.

**Methodology:** Jacqueline Ntombizodwa Ndlovu, Jura L. Augustinavicius, Wietse A. Tol.

**Project administration:** Lena S. Andersen, Marx R. Leku, Nawaraj Upadhaya, Morten Skovdal, Jura L. Augustinavicius, Wietse A. Tol.

**Resources:** Nawaraj Upadhaya, Jura L. Augustinavicius, Wietse A. Tol.

**Supervision:** Nawaraj Upadhaya, Morten Skovdal, Jura L. Augustinavicius, Wietse A. Tol.

**Writing – original draft:** Jacqueline Ntombizodwa Ndlovu.

**Writing – review & editing:** Jacqueline Ntombizodwa Ndlovu, Lena S. Andersen, Marx R. Leku, Nawaraj Upadhaya, Morten Skovdal, Jura L. Augustinavicius, Wietse A. Tol.

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
