## [Decision Letter · Decision Letter 0]

30 Jul 2024

PMEN-D-24-00226

“My home is (now) at peace”: Evaluating the relevance, acceptability and potential scalability of a guided self-help intervention for male refugees in Uganda

PLOS Mental Health

Dear Dr. Ndlovu,

Thank you for submitting your manuscript to PLOS Mental Health. After careful consideration, we feel that it has merit but does not fully meet PLOS Mental Health’s publication criteria as it currently stands. Therefore, we invite you to submit a revised version of the manuscript that addresses the points raised during the review process.

Please note that we have only been able to secure a single reviewer to assess your manuscript. We are issuing a decision on your manuscript at this point to prevent further delays in the evaluation of your manuscript. Please be aware that the editor who handles your revised manuscript might find it necessary to invite additional reviewers to assess this work once the revised manuscript is submitted. However, we will aim to proceed on the basis of this single review if possible. 

We look forward to receiving your revised manuscript.

Kind regards,

Vanessa Carels

Staff Editor

PLOS Mental Health

Journal Requirements:

Additional Editor Comments (if provided):

Reviewers' comments:

Reviewer's Responses to Questions

**Comments to the Author**

1. Does this manuscript meet PLOS Mental Health’s publication criteria? Is the manuscript technically sound, and do the data support the conclusions? The manuscript must describe methodologically and ethically rigorous research with conclusions that are appropriately drawn based on the data presented.

Reviewer #1: Yes

2. Has the statistical analysis been performed appropriately and rigorously?

Reviewer #1: N/A

3. Have the authors made all data underlying the findings in their manuscript fully available (please refer to the Data Availability Statement at the start of the manuscript PDF file)?

Reviewer #1: No

4. Is the manuscript presented in an intelligible fashion and written in standard English?

Reviewer #1: Yes

5. Review Comments to the Author

Reviewer #1: Reviewer’s Report

Overall, this was a very interesting and relevant paper. However, there is an opportunity for the authors to further distinguish their findings by contextualizing the study in a cultural context earlier on. Despite all of the participants being of South Sudanese descent, no specific mention of cultural nuances was made until the discussion section. More information on cultural dynamics, cultural differences between refugees and the host community, and displacement context will help establish a better understanding from readers who may be unfamiliar with the adversities faced by this specific refugee population in this particular host country.

Introduction

Line 51-54: gender disparities in psychological distress and substance misuse are mentioned without a description of the specific factors unique to male refugees that worsen these outcomes.

Line 56-59: the prevalence of alcohol misuse among male refugees is reported, but could use more context as to why this is a coping mechanism often used by this population.

Line 60-61: the relationship between alcohol misuse and mental health needs to be more clearly established. The statistics leading up to this claim only report the prevalence of each concern independently.

Line 63-68: the connection between gaps in mental health resources and access to actual intervention content is unclear.

Line 72-75: stigma and discrimination are noted only in the introduction and not circled back in the results/discussion.

Methods

Why was the study conducted in the two refugee settlements in northern Uganda? The introduction mentioned limited evidence on alcohol misuse and mental health challenges among refugees in low and middle-income countries. Was there any other reason besides the host country's economic status?

Was there any specific reasoning behind the inclusion criteria related to the participants? Or was this just a matter of a majority of refugees in these settlements participating in this program being South Sudanese males?

181 participants met the inclusion criteria but only 18 participated in the process evaluations (28 participants including family and facilitators). Was this drop in participants solely due to availability and willingness? Were all 181 participants presented with the opportunity to participate or was there a cap due to resource limitations?

Line 162-163: note that scores greater than or equal to 5 were classified as moderate psychological distress based on “previous studies with South Sudanese refugees” but only cite one empirical study. Based on the original instrument, scores of 8-12 indicated moderate distress.

Line 190: provide examples of task shifting. What does that entail and how could this potentially impact experiences across participants?

Line 197-199: Was the combined intervention validated on the study's population at any point during this study? If this study tries to fill the gap in knowledge by bringing interventions targeting two related but separate outcomes, these might need to be for future research directions or so.

Line 227: What kind of adaptations to the materials were made based on the feedback collected from intervention facilitators?

Line 243: Who were the intervention facilitators and how many were there? What was the ratio of facilitators to participants across all of the different interventions?

Results

Table 1 is a bit confusing, consider reformatting. For example, why is the percentage of participants in each intervention group in the context of total participants if there were only 18 participants and the rest of the sample were family/facilitators?

The section on acceptability of the intervention doesn’t reference relevant content based on the earlier definition of acceptability referring to “satisfaction with the intervention components, content, delivery” (Line 114-115). The content in this section seems more aligned with the author's definition of relevance (Line 113-114), “the appropriateness, practicability and perceived usefulness of the intervention”.

Line 559-563: mentioned other stakeholders and the importance of those relationships but who are the stakeholders and what was their role in this study? How did these individuals ease the mobilization of participants? Were there any additional impacts?

Discussion

Line 693-694: It was indicated that more participants from the combined intervention participated in the process evaluation (over 40%). This should be noted earlier and more clearly in the results so readers can consider that when reviewing the main outcomes.

New concepts such as livelihood interventions (lines 638-650) and gender roles (lines 652-671) are introduced at the end and while they are interesting they don’t seem to be directly related to the original study goals. Consider integrating this literature earlier on in the background section, particularly the content on gender norms.

6. PLOS authors have the option to publish the peer review history of their article (what does this mean?). If published, this will include your full peer review and any attached files.

**Do you want your identity to be public for this peer review?** For information about this choice, including consent withdrawal, please see our Privacy Policy.

Reviewer #1: No

---

## [Decision Letter · Decision Letter 1]

15 Jan 2025

“My home is (now) at peace”: Evaluating the relevance, acceptability and potential scalability of a guided self-help intervention for male refugees in Uganda

PMEN-D-24-00226R1

Dear Ms Ndlovu,

We are pleased to inform you that your manuscript '“My home is (now) at peace”: Evaluating the relevance, acceptability and potential scalability of a guided self-help intervention for male refugees in Uganda' has been provisionally accepted for publication in PLOS Mental Health. During the second assessment of the paper, we brought in a second reviewer as per our policy, and they closely assessed the paper and the revisions made. They found the work to be solid and the revisions sufficient and so we are now comfortable proceeding with this paper having had two external reviews.

Best regards,

Karli Montague-Cardoso

Executive Editor

PLOS Mental Health

Reviewer Comments (if any, and for reference):

Reviewer's Responses to Questions

**Comments to the Author**

1. If the authors have adequately addressed your comments raised in a previous round of review and you feel that this manuscript is now acceptable for publication, you may indicate that here to bypass the “Comments to the Author” section, enter your conflict of interest statement in the “Confidential to Editor” section, and submit your "Accept" recommendation.

Reviewer #2: All comments have been addressed

2. Does this manuscript meet PLOS Mental Health’s publication criteria? Is the manuscript technically sound, and do the data support the conclusions? The manuscript must describe methodologically and ethically rigorous research with conclusions that are appropriately drawn based on the data presented.

Reviewer #2: Yes

3. Has the statistical analysis been performed appropriately and rigorously?

Reviewer #2: Yes

4. Have the authors made all data underlying the findings in their manuscript fully available (please refer to the Data Availability Statement at the start of the manuscript PDF file)?

Reviewer #2: Yes

5. Is the manuscript presented in an intelligible fashion and written in standard English?

Reviewer #2: Yes

6. Review Comments to the Author

Reviewer #2: The work of Tol and colleagues evaluates the guided self-help intervention developed by this highly experienced team. It addresses psychological distress caused by severe and traumatic stress, including war and poverty, in the current studies with a focus on alcohol misuse. This paper seeks to determine acceptability and potential scalability. This is of paramount importance given the large number of refugees which worldwide ranges above 100 million.

The research was conducted in Uganda's West Nile refugee settlements and included 28 in-depth process evaluation interviews with participants, their family members and intervention facilitators. The findings reaffirm the need for and relevance of scalable intervention in war-displaced populations. Acceptance was underlined by a sense of community and social acceptance. Facilitators noted that the format of the intervention was key to scalability, despite barriers such as competing priorities and limited resources. Given the prevalence of alcohol misuse among displaced populations, the focus on alcohol misuse is particularly important. However, the question remains as to how much subjective information from alcohol dependent men can be trusted. In addition, the fact that only 18 of the 181 study participants took part in the process evaluations (28 participants including family members and facilitators) further weakens the results. The authors have addressed this last point in their response.

Overall, the paper is very well written and concise. The introduction convincingly summarises the importance of addressing alcohol misuse and trauma-related distress in parallel. The method gives a description of the subjects and procedures that is sufficiently detailed for others to be able to replicate the investigations. Overall, the analysis was appropriate. Results are presented in detail. The discussion relates to the main findings..

In conclusion, the authors have successfully revised their manuscript which offers an important contribution to the field.

7. PLOS authors have the option to publish the peer review history of their article (what does this mean?). If published, this will include your full peer review and any attached files.

**Do you want your identity to be public for this peer review?** For information about this choice, including consent withdrawal, please see our Privacy Policy.

Reviewer #2: **Yes: **Thomas Elbert
